# Fooling Explanations in Text Classifiers

**Adam Ivankay**
IBM Research Zurich
Rüschlikon, Switzerland
aiv@zurich.ibm.com

**Ivan Girardi**
IBM Research Zurich
Rüschlikon, Switzerland
ivg@zurich.ibm.com

**Chiara Marchiori**
IBM Research Zurich
Rüschlikon, Switzerland
chi@zurich.ibm.com

**Pascal Frossard**
École Polytechnique Fédérale de Lausanne (EPFL)
Lausanne, Switzerland
pascal.frossard@epfl.ch

## Abstract

State-of-the-art text classification models are becoming increasingly reliant on deep neural networks (DNNs). Due to their black-box nature, faithful and robust explanation methods need to accompany classifiers for deployment in real-life scenarios. However, it has been shown in vision applications that explanation methods are susceptible to local, imperceptible perturbations that can significantly alter the explanations without changing the predicted classes. We show here that the existence of such perturbations extends to text classifiers as well. Specifically, we introduce TextExplanationFooler (TEF), a novel explanation attack algorithm that alters text input samples imperceptibly so that the outcome of widely-used explanation methods changes considerably while leaving classifier predictions unchanged. We evaluate the performance of the attribution robustness estimation performance in TEF on five sequence classification datasets, utilizing three DNN architectures and three transformer architectures for each dataset. TEF can significantly decrease the correlation between unchanged and perturbed input attributions, which shows that all models and explanation methods are susceptible to TEF perturbations. Moreover, we evaluate how the perturbations transfer to other model architectures and attribution methods, and show that TEF perturbations are also effective in scenarios where the target model and explanation method are unknown. Finally, we introduce a *semi-universal* attack that is able to compute fast, computationally light perturbations with no knowledge of the attacked classifier nor explanation method. Overall, our work shows that explanations in text classifiers are very fragile and users need to carefully address their robustness before relying on them in critical applications.

## 1 Introduction

Deep neural networks (DNNs) have undoubtedly become the state-of-the-art architectures for many existing machine learning tasks (Choi et al., 2016). Yet, their *black-box* nature has raised the need for developing methods to mitigate the lack of interpretability caused by their increased complexity (Simonyan et al., 2013; Zeiler & Fergus, 2014; Hendricks et al., 2018; Bahdanau et al., 2014). These methods give intuitive, easily understandable explanations that do not require significant domain knowledge. This is especially desired in safety-critical scenarios, such as healthcare, where the users of such DNNs - the medical professionals for instance - need to understand the decision process and reasoning behind it. However, they have been shown to lack local robustness towards carefully crafted, imperceptible perturbations in the input (Ghorbani et al., 2019). While resulting in the same predictions, these altered inputs yield significantly different explanations and attributions maps (Figure 1). Interpretation methods fragile towards small input perturbations not only fail to provide faithful explanations, a desiderata commonly required in explainable AI (Jacovi & Goldberg, 2020), but also damages user trust in DNNs and prevents them from being deployed on high-stakes, safety-

| Original sample | TEF sample | PCC |
|---|---|---|
| [CLS] romanians pitch **rumsfeld** on base location — mihail **kogalniceanu** air base , **romania** - to entice the us military to make a home here , what better symbolic appeal could the romanian government make than to rename a street here quot;george washington boulevard ? [SEP]

$F(s, \mathbf{World}) = 0.99$ | [CLS] **romanians** pitch clinton on base places — **mihail kogalniceanu** air base , **rumania** - to entice the us military to make a home here , what better symbolic appeal could the romanian government make than to rename a street here quot;george washington **boulevard** ? [SEP]

$F(s, \mathbf{World}) = 0.97$ | -0.07 |
| [CLS] **forgettable horror –** more **gory** than psychological **–** with a highly **satisfying quotient of friday** - night **excitement and** milla **power .** [SEP]

$F(\boldsymbol{s}, \mathbf{Pos}.) = 0.99$ | [CLS] forgettable horror – more gory than psychological – with a **highly satisfying** quotient **of** friday - night **arousal and milla** wattage . [SEP]

$F(\boldsymbol{s}, \mathbf{Pos}.) = 0.99$ | 0.18 |

Figure 1: Example of fragile attributions. Highlighted red words are deemed most important *towards* the predicted class by the Integrated Gradients attribution method, blue ones *against* it. By substituting a few words in the original sample, the *Pearson Correlation Coefficient* (PCC) of word importances drops to below 0.2 while maintaining the same confidence in the correctly predicted class (denoted by $F$).

critical applications, such as in healthcare (Adadi & Berrada, 2020).

The previously described phenomenon has been widely studied in the image domain by Etmann et al. (2019); Moosavi-Dezfooli et al. (2019b) or Ivankay et al. (2020). However, in discrete-input domains like text, there has been limited progress. This is especially problematic given the increased reliance on and fragility of attention mechanisms as *inherently explainable* methods, as stated in Ghaeini et al. (2018). Therefore, we summarize our contributions as follows:

- We provide a novel baseline black-box adversarial attack, TEXTEXPLANATIONFOOLER (TEF) to estimate the local robustness of *explanations* in text classification problems

- We evaluate attribution robustness on widely used, state-of-the-art text datasets and model architectures, showing that explanation methods' output can be significantly altered with our attack

- We provide insight into the *transfer capability* of TEF on different models and explanation methods as well as introduce *semi-universal* adversarial perturbations to alter explanations without requiring access to the model at attack-time

## 2 PRELIMINARIES

### 2.1 RELATED WORK

Adversarial attacks that alter the inference outcomes in DNNs have been widely studied both in the image (Goodfellow et al., 2014; Carlini & Wagner, 2017; Moosavi-Dezfooli et al., 2016; Modas et al., 2019) and text domain (Ebrahimi et al., 2017; Sun et al., 2020; Jin et al., 2019; Yang et al., 2020). Methods to alleviate the networks susceptibility to such attacks have also been proposed, including the works of Madry et al. (2017); Moosavi-Dezfooli et al. (2019a); Buckman et al. (2018) or Cisse et al. (2017). However, it has recently been shown by authors Ghorbani et al. (2019) that, in addition to DNN predictions, widely-used *explanation methods* also lack robustness to targeted, imperceptible alterations of the input. These attacks change the outcomes of such explanation methods significantly, while predictions of the DNNs are unaltered. This violates the *Prediction Assumption* of *faithful* explanations and crucially degrades user trust in such explanation methods, as significantly different interpretations are provided for similar inputs and outputs (Jacovi & Goldberg, 2020). The aforementioned phenomenon of fragile explanations has mostly been investigated in the image domain (Etmann et al., 2019; Singh et al., 2019; Chen et al., 2019; Ivankay et al., 2020), with less focus on discrete input spaces like text. However, faithful and robust interpretations are arguably equally important in the discrete text domain, for instance in electronic health record classification

(Girardi et al., 2018) or precision medicine (Binder et al., 2021), where critical decisions often need to be based on DNN explanations. The work of La Malfa et al. (2021) constructs inherently robust explanations for NLP models, however only towards perturbations in the embedding space, not the input space that adversaries can operate on. Moreover, they do not give a method to evaluate robustness of already existing explanation algorithms. The authors of Feng et al. (2018) show that interpretation methods in NLP lack completeness (Sundararajan et al., 2017) by removing words deemed least important by explanation methods. Moreover, attention mechanisms (Bahdanau et al., 2014) have been increasingly relied on as *inherently* interpretable systems. However recent work has questioned their faithfulness and plausibility (Serrano & Smith, 2019; Jain & Wallace, 2019; Wiegreffe & Pinter, 2019) by proving that they often do not highlight input components that are most important to a DNN decision. We, paralleled by the very recent work of Sinha et al. (2021), are the first to show that imperceptible perturbations in the *input space* can alter the outcome of explanations of text classifiers significantly, giving an efficient attack to estimate explanation robustness. However, our work is the first to give an extensive evaluation of the robustness of widely-used explanation methods on large datasets, comparing several state-of-the-art architectures. Further, we are the first to address robustness of attention weights in transformer architectures and to provide insight on transfer capabilities and universal attacks.

## 2.2 BACKGROUND

Let $\mathbb{S} = \{s_1, s_2, ..., s_N\}$ be a dataset of $N$ text samples $s_i$, each with a label from a predefined set of labels $\mathbb{L} = \{l_1, l_2, ..., l_{|\mathbb{L}|}\}$. Each sample $s_i$ contains a sequence of tokens (or words) $w_i$ taken from a discrete vocabulary set $\mathbb{W} = \{w_1, w_2, ..., w_{|\mathbb{W}|}\}$. A generic sequence classifier then consist of a non-injective, non-surjective embedding function $E : \mathbb{S} \to \mathbb{R}^{d \times p}$, $E(s) = X$, which maps the input sample $s$ to its embedding matrix $X$, and a function $f : \mathbb{R}^{d \times p} \to \mathbb{R}^{|\mathbb{L}|}$, $f(X) = o$, representing a (DNN) classifier function. $d$ and $p$ denote the embedding dimension and sequence length respectively. Let $F : \mathbb{S} \to \mathbb{R}^{|\mathbb{L}|}$, $F(s) = f \circ E$ be the full sequence classifier with final prediction $y = \arg\max_{i \in \{1:|L|\}} o_i$.

We define an attribution map as $A : \mathbb{S} \to \mathbb{R}^p$, $A(s, F, l) = a$ that assigns a scalar value to each input token $w_i$ in the text sample $s$, resulting in the attribution vector $a \in \mathbb{R}^p$. This vector represents each token's influence towards the prediction outcome $y$ of classifier $F$. Our work considers three widely-used attribution methods in text classification, namely Saliency Maps (S) (Simonyan et al., 2013), Integrated Gradients (IG) (Sundararajan et al., 2017) and Attention (A) (Bahdanau et al., 2014), defined in the following Equations (1), (2) and (3) respectively.

$$A_i^{\text{S}}(s, F, l) = \sum_{j \in \{1:d\}} |\nabla_X f(X)_l|_{j,i} \tag{1}$$

$$A_i^{\text{IG}}(s, F, l, B) = \sum_{j \in \{1:d\}} \left[ (X - B) \cdot \int_{\alpha=0}^1 \nabla_{\tilde{X}} f(\tilde{X})_l |_{\tilde{X}=B+\alpha(X-B)} \, d\alpha \right]_{j,i} \tag{2}$$

$$A_i^{\text{Att}}(s, F, l) = \frac{\exp e_i}{\sum_{j \in \{1:p\}} \exp e_j} \tag{3}$$

where $B$ denotes the null matrix $\mathbf{0}^{d \times p}$, $f$ is the classifier function of $F$, $\alpha$ a scaling factor and $X = E(s)$. $\nabla_X f$ denotes the matrix-derivative of $f$ to $X$, as defined in Goodfellow et al. (2016). An attention head is a layer that transforms its inputs into scores $e$ and calculates its output by linear combination of each input score, with coefficients normalized to a distribution. These coefficients are the attention weights $A_i^{\text{Att}}(s, F, l)$ denoted in Equation (3). It is commonly agreed to give intuitive explanations on how much the model *attends* to the given inputs through its attention weights (Jacovi & Goldberg, 2020).

## 3 METHODS

In this section, we describe our novel method TEXTEXPLANATIONFOOLER (TEF) to estimate attribution robustness (AR) in sequence classification problems. Specifically, we define the problem formulation, introduce our threat model and present the algorithm used to alter explanations by imperceptibly changing the inputs.

## 3.1 PROBLEM FORMULATION

Given an input text samples $\boldsymbol{s}$ and $\tilde{\boldsymbol{s}}$, labels $l$; a text classifier $F$ with embedding function $E$ and classifier function $f$; and attribution method $A$, we define **attribution robustness** (also *explanation robustness*, AR) as written in Equation (4).

$$r(\tilde{\boldsymbol{a}}, \boldsymbol{a}) = 1 - \max_{\tilde{\boldsymbol{a}}} \, d(\tilde{\boldsymbol{a}}, \boldsymbol{a}) = 1 - \max_{\tilde{\boldsymbol{s}}} \, d\big[A(\tilde{\boldsymbol{s}}, F, l), \; A(\boldsymbol{s}, F, l)\big] \tag{4}$$

with

$$\arg\max_{i \in \{1:|\mathbb{L}|\}} F(\tilde{\boldsymbol{s}}) = \arg\max_{i \in \{1:|\mathbb{L}|\}} F(\boldsymbol{s}), \tag{5}$$

where $d$ denotes a distance measure between the attributions $\tilde{\boldsymbol{a}}$ and $\boldsymbol{a}$ of the the two input samples $\boldsymbol{s}$ and $\tilde{\boldsymbol{s}}$. The rest of the notation is kept as in Section 2. Equation (4) quantifies how different the attributions of two input samples are, given the constraint in Equation (5) that enforces the inputs having the same prediction outcome.

The attribution robustness estimation is then solved utilizing the following Equation (6).

$$\boldsymbol{s}_{\text{adv}} = \arg\max_{\tilde{\boldsymbol{s}}} \, d\big[A(\tilde{\boldsymbol{s}}, F, l), \; A(\boldsymbol{s}, F, l)\big] \tag{6}$$

where $\boldsymbol{s}_{\text{adv}}$ denotes the solution to the estimation, i.e. the adversarial input, which also minimizes AR defined in Equation (4). $\boldsymbol{s}$ denotes the original, unperturbed input and $\tilde{\boldsymbol{s}}$ the perturbed input, optimized during estimation. The solution $\boldsymbol{s}_{\text{adv}}$ gives a robustness estimate by finding an input that maximizes the distance between original attribution $A(\boldsymbol{s}, F, l)$ and adversarial attribution $A(\tilde{\boldsymbol{s}}, F, l)$ within a local neighbourhood of $\boldsymbol{s}$. The more *dissimilar* these maps are, the less robust the attribution method is. The local neighbourhood is defined by both linguistic constraints described in the next section that encourage semantic proximity to the original text and the perturbed samples having the same prediction outcome as the unperturbed ones, see Equation (5). This formulation is backed by current research (Ghorbani et al., 2019; Dombrowski et al., 2019; Ivankay et al., 2020) and the *Prediction Assumption* of faithful explanations (Jacovi & Goldberg, 2020).

## 3.2 THREAT MODEL AND ATTACK

We define our algorithm to estimate AR as a *black-box* attack. It only queries the model to obtain its output logits and the accompanied explanations of the inference process. The model might access its gradients to compute explanations, but the attack only utilizes the resulting explanations, no gradient or architectural information. We restrict the valid input perturbations to token substitutions, specifically insertions and deletions of tokens are forbidden, as they alter the input lengths. Algorithm 1 contains the schematic code for TEF, consisting of the following two steps.

**Step 1 - Word importance ranking** First, an importance ranking is extracted for each token of the input sample. Specifically, we compute $I_{w_i} = d\big[A(\boldsymbol{s}_{w_i \to 0}, F, l), \; A(\boldsymbol{s}, F, l)\big]$ for each token $i$ in $\boldsymbol{s}$, where $\boldsymbol{s}_{w_i \to 0}$ denotes the input sequence $\boldsymbol{s}$ with the $i$-th word masked to the zero embedding token. The input tokens are then sorted by the $I_{w_i}$ values in a decreasing fashion. Then, high importance words are prioritized during substitution.

---

**Algorithm 1** `TextExplanationFooler` (TEF)

---

**Input**: Input sentence $\boldsymbol{s}$ with predicted class $l$, classifier $F$, attribution $A$, attribution distance $d$, number of synonyms $N$, maximum perturbation ratio $\rho_{max}$

**Output**: Adversarial sentence $\boldsymbol{s}_{\text{adv}}$

1: $\boldsymbol{s}_{\text{adv}} \leftarrow \boldsymbol{s}, d_{max} \leftarrow 0, r \leftarrow 0$
2: **for** $w_i \in \boldsymbol{s}$ **do**
3:      $I_{w_i} = d\big[A(\boldsymbol{s}_{w_i \to 0}, F, l), \; A(\boldsymbol{s}, F, l)\big]$
4: **for** $w_j \in \langle w_1, ..., w_{|\boldsymbol{s}|} \rangle \mid I_{w_{m-1}} \geq I_{w_m} \; \forall m \in \{2, ..., |\boldsymbol{s}|\}$ **do**
5:      **if** $w_j \in \mathbb{S}_{\text{Stop words}}$ **then**
6:          **continue**
7:      $\mathbb{C}_j \leftarrow \text{SynonymEmbeddings}(w_j, N)$
8:      $\mathbb{C}_j \leftarrow \text{POSFilter}(w_j, \mathbb{C}_j, \boldsymbol{s})$
9:      **for** $c_k \in \mathbb{C}_j$ **do**
10:          $\tilde{\boldsymbol{s}}_{w_j \to c_k} \leftarrow$ Replace $w_j$ in $\boldsymbol{s}_{\text{adv}}$ with $c_k$
11:          **if** $\arg\max_{i \in \{1:|\mathbb{L}|\}} F(\tilde{\boldsymbol{s}}_{w_j \to c_k}) = l$ **then**
12:              $\tilde{d} \leftarrow d\big[A(\tilde{\boldsymbol{s}}_{w_i \to c_k}, F, l), \; A(\boldsymbol{s}, F, l)\big]$
13:              **if** $\tilde{d} > d_{max}$ **then**
14:                  $\boldsymbol{s}_{\text{adv}} \leftarrow \tilde{\boldsymbol{s}}_{w_i \to c_k}$
15:                  $d_{max} \leftarrow \tilde{d}$
16:                  $r \leftarrow r + 1$
17:      **if** $\rho = \frac{r}{|\boldsymbol{s}|} + 1 > \rho_{max}$ **then**
18:          **break**

---

Table 1: Accuracies, average text length and number of classes of our models trained on the five datasets.

| DATASET | CNN | LSTM | LSTMATT | BERT | RoBERTA | XLNET | Mean $|s|$ | $|\mathbb{L}|$ |
|---|---|---|---|---|---|---|---|---|
| AG's NEWS | 89.7% | 90.8% | 91.4% | 94.2% | 94.0% | 93.8% | 45 | 4 |
| IMDB | 82.0% | 87.2% | 87.3% | 89.4% | 93.3% | 93.7% | 270 | 2 |
| FAKE NEWS | 98.9% | 99.6% | 99.6% | 99.8% | 100.0% | 100.0% | 919 | 2 |
| MR | 73.0% | 76.4% | 78.0% | 82.2% | 87.7% | 86.3% | 22 | 2 |
| YELP | 49.0% | 54.8% | 60.0% | 62.6% | 67.6% | - | 159 | 5 |

**Step 2 - Candidate selection**    For each word $w_i$ in $s$ sequentially, a set of *substitution candidates* $\mathbb{C}$ of $N$ elements is extracted. This candidate set is constructed from the counter-fitted GloVe (Pennington et al., 2014) synonym embeddings by the authors of Mrkšić et al. (2016). The candidates are then filtered by Part-Of-Speech (POS), tagged by SpaCy (Honnibal et al., 2020), only allowing replacements with equal POS. Stop words are also discarded from $\mathbb{C}$. A *final selection* as replacement for $w_i$ is then made to be the $c_k \in \mathbb{C}$ that maximizes $d\big[A(\tilde{s}_{w_i \to c_k}, F, l), \ A(s, F, l)\big]$. The algorithm is aborted when the number of replacements to sentence length exceeds the maximum value $\rho_{max}$.

## 4 EXPERIMENTS AND RESULTS

In this section, we present an extensive evaluation of our attribution robustness (AR) estimation attack, TEF, for sequence classification problems. We examine the performance of TEF and study the impact of different factors on its robustness evaluation performance. We find that our attack effectively reduces the correlation of original and attacked attributions on all datasets and models. Moreover, we describe our transfer and semi-universal attacks and examine their robustness estimation performance, showing that even under circumstances where the model and explainer are unknown to the attacker, TEF perturbations transferred from other models decrease attribution robustness effectively.

### 4.1 MODELS, DATASETS AND EVALUATION

Our TEF attack is evaluated on five commonly used public sequence classification datasets, AG's News (Zhang et al., 2015), MR reviews (Zhang et al., 2015), IMDB Movie Reviews (Maas et al., 2011), Fake News Dataset [1] and Yelp (Asghar, 2016). We train six different word embedding-based architectures for each dataset, namely a CNN, an LSTM, an LSTM containing a single attention layer with one head (LSTMAtt) and three state-of-the-art finetuned transformer-based architectures, BERT (Li et al., 2020), RoBERTa (Liu et al., 2019) and XLNet (Yang et al., 2019). Table 1 contains a summary of our model performances as well as details on the datasets. The text samples are tokenized with the default English SpaCy (Honnibal et al., 2020) tokenizer for the CNN, LSTM and LSTMAtt models and embedded with the pretrained GloVe 6B 300-dimensional word vectors (Pennington et al., 2014). The transformer-based models use their own pretrained tokenizers and embeddings. We use PyTorch (Paszke et al., 2019) with Captum (Kokhlikyan et al., 2020) to implement our models and explainers and the Huggingface Transformers library (Wolf et al., 2020) to finetune the transformer architectures on our datasets.

We evaluate the robustness of three commonly used explanation methods in natural language processing with our TEF attack. These are Saliency Maps (S), Integrated Gradients (IG) and the Attention mechanism (A), defined in Section 2. We use S and IG in combination with all our architectures, Attention only with LSTMAtt, BERT, RoBERTa and XLNet. During the attack, we set the attribution distance $d$ of Equation (6) to be $d(\tilde{a}, a) = 1 - \dfrac{\text{PCC}(\tilde{a}, a) + 1}{2}$, with PCC denoting the Pearson Correlation Coefficient (Pearson, 1895) of original and adversarial attributions $\tilde{a}$ and $a$. We then report the standard Pearson Correlation Coefficient (PCC), Kendall's Rank Order Correlation (ROC) (Kendall, 1938), Spearman's Correlation Coefficient (SCC) (Myers & Sirois, 2004) and the Top-10%, Top-30% and Top-50% intersections to measure AR in Equation (4). These are common

---

[1]https://www.kaggle.com/c/fake-news/data

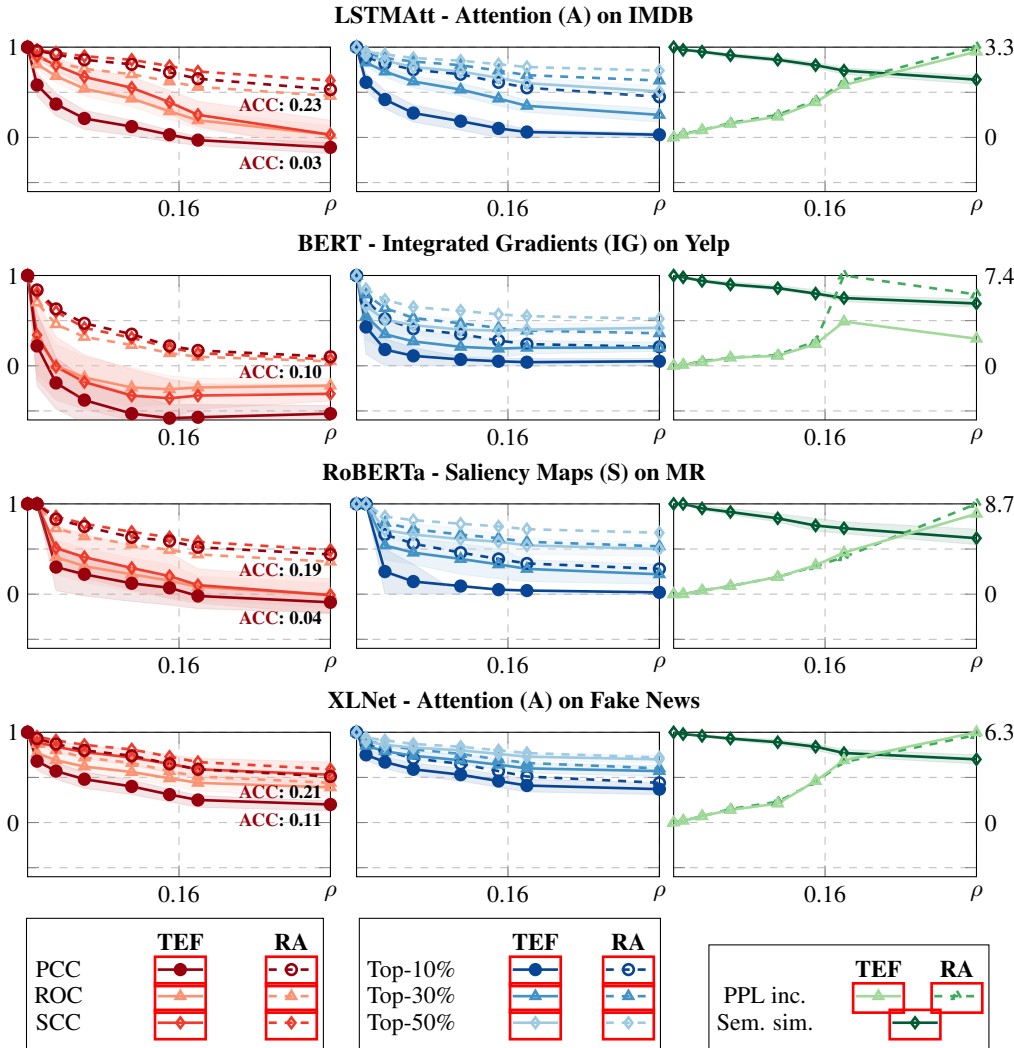

Figure 2: Robustness of attribution maps on several architectures and explainers. We plot the average correlations (PCC, ROC, SCC) (left), the Top-10%, Top-30% and Top-50% intersections (middle), the semantic similarity and increase of average perplexity (right) as functions of the perturbed ratio $\rho$. Dashed lines indicate the metrics for our RANDOMATTACK (RA). The ACC indicates the area under the PCC ( —●— and - ⊙ - ) curve, lower values correspond to overall lower feature attribution correlations in the overall operation interval of $\rho$. The perplexity increases are indicated on the right axis, all other metrics on the left.

metrics that correspond to human measures of AR (Ghorbani et al., 2019; Dombrowski et al., 2019; Ivankay et al., 2020). Additionally, in order to quantify imperceptibility of perturbations, the *semantic similarity* of adversarially perturbed and unchanged sentences is reported, along with the relative increase of average *perplexity* of the perturbed samples, given by the GPT-2 (Radford et al., 2019) language model. Semantic similarity (Sem. sim.) is measured by the cosine distance between the embeddings produced by the Universal Sentence Encoder (USE) (Cer et al., 2018). This is a state-of-the-art sentence embedding widely used in adversarial attacks on text (Sun et al., 2020; Jin et al., 2019). Perplexity increase (PPL inc.) indicates how much the *likelihood* of the perturbed data has decreased, given a language model, and is often used to validate language models (Keselj, 2009).

Due to the lack of related work in this field, we compare the AR estimation performance of TEF to our RANDOMATTACK (RA) baseline. RA serves as an agnostic attack, utilizes a random word importance ranking in Step 1 of TEF and selects a random synonym in the *final selection* in Step 2. POS and stop word filters (see Section 3) are still utilized in RA to keep linguistic constraints intact.

**PCC of LSTMAtt - A on AG's News**   **PCC of LSTM - IG on MR**

Figure 3: Ablation study of TEF. We compare the PCC of TEF, RA, the RANDOMIMPORTANCE (RI) attack and the RANDOMSYNONYM (RS) attack. We find that RI behaves slightly worse than TEF, while RS behaves slightly better than RA in terms of reducing attribution correlation over all $\rho$ values.

## 4.2 ROBUSTNESS OF EXPLANATIONS

**Attribution robustness estimation.** In order to estimate the attribution robustness (AR) of the aforementioned models and explainers, we vary the parameter $\rho_{max}$ of TEF, which denotes the maximum ratio of perturbed tokens in the input sample. A larger $\rho_{max}$ value leads to lower attribution correlation, as potentially more words are substituted in the input. We then capture the aforementioned metrics PCC, ROC, SCC, Sem. sim, Top-10%/30%/50% intersections and PPL inc. to evaluate AR. Additionally, in order to quantify performance of our attack over the whole operation interval of $0 \leq \rho_{max} \leq 0.4$, we compute the Area under the Pearson Correlation Curve (ACC). A lower value of ACC corresponds to lower robustness overall, as correlation values are lower. We note that a particular value of $\rho_{max}$ does not guarantee that all input samples have exactly $\rho_{max}$ ratio of perturbed tokens. Therefore, we quantize our samples based on their actual, resulting perturbed ratio $\rho$ such that samples with similar $\rho$ are grouped together. These bins are computed per dataset, ensuring the comparability of resulting curves and ACCs for each plot. Moreover, we choose the number of candidates in Step 2 of TEF to be $N = |\mathbb{C}| = 15$, as it is a good trade-off between TEF estimation performance and attack run time. As expected, we find that TEF is able to significantly outperform the baseline provided by RA in terms of all AR metrics, on all datasets, models and explanation methods considered in this work. A subset of these results is shown in Figure 2, the rest can be found in Appendix A.2. Moreover, we do not find that any architecture is significantly more robust to TEF perturbations for explainers S and IG. However, the self-attention mechanism of transformers seems to be more robust to perturbations than non-transformer-based architectures and explanations. The semantic similarity decreases with increasing $\rho$ and stays above 0.7 in most cases. This, together with the fact that resulting samples share predictions with the non-perturbed ones effectively highlights that the explanations given by these models and attribution methods lack *faithfulness*.

**Ablation study.** In addition to the fully random attack described in the previous paragraph, we compare TEF to our semi-random attacks RANDOMIMPORTANCE (RI) and RANDOMSYNONYM (RS). We randomize the word importance ranking of TEF (RI) but keep the selection of best final synonym, and we randomize the final synonym selection of TEF (RS) but keep the word importance ranking respectively. Figure 3 shows our findings for these experiments, along with comparisons to RANDOMATTACK (RA). The PCC curves and the ACC values show that RI consistently outperforms RS in terms of PCC over the whole operation interval of $\rho$. Moreover, the impact of word importance ranking diminishes with increasing $\rho$, especially for shorter datasets like MR. This can be observed by RS performing closer to RA for high $\rho$ values.

**BERT's attention layers and heads.** BERT's attention weights can be used to help gain insight into a models prediction by understanding which parts of the input are most *attended* to (Vig, 2019). Our BERT models have 12 layers with 12 attention heads (144 heads in total), each producing a distribution of attention weights over its inputs and outputs. Estimating the AR of all heads together is not useful, as effects would average out. Therefore, we run TEF to estimate the robustness of each head separately. Figure 4 contains the average PCCs of the attention weights before and after perturbing the inputs with TEF. We find that attention weights in later layers tend to be more susceptible to input perturbations than earlier layers. Moreover, heads within a layer tend to be comparably robust. We leave a thorough, theoretical analysis of this phenomenon to future work. We conclude that

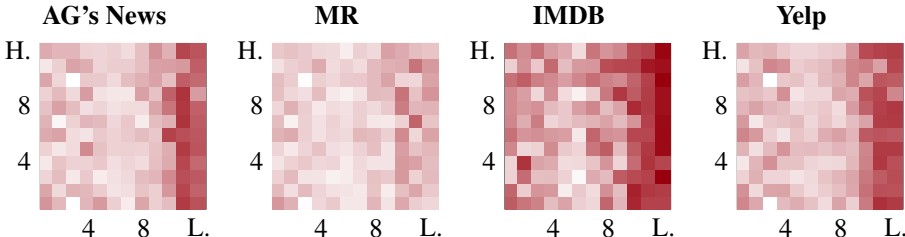

Figure 4: Estimated robustness of BERT attention weights on different layers (X-axis) and heads (Y-axis) for $\rho_{max} = 0.2$. Red cells indicate average PCC values close to 0, hence less robust attention head weights, while white cells have average PCCs close to 1. Attention heads in later layers tend to be less robust, while heads within a layer seem equally robust in most layers.

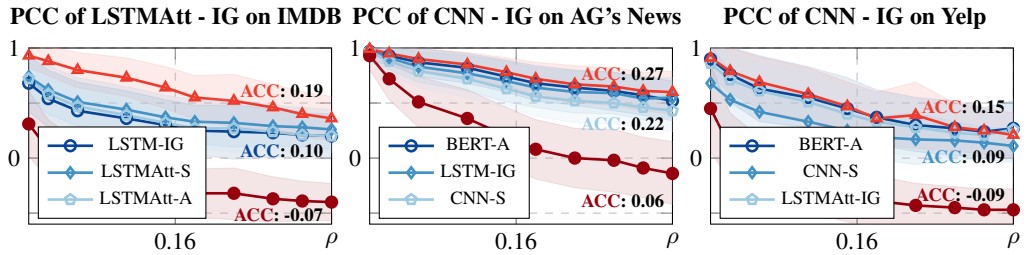

Figure 5: Transfer capabilities of TEF to other models and explanation methods. The lines indicate the estimated PCC of TEF perturbations transferred from the indicated models and explanations. ●—● and ▲—▲ indicate the PCC curve of optimal TEF and RA perturbations respectively, without transfer.

the increasing reliance on attention weights to provide inherent interpretations to BERT predictions needs careful investigation, especially in safety-critical applications.

### 4.3 TRANSFERABILITY AND SEMI-UNIVERSAL PERTURBATIONS

**Transferability of perturbations to models and explanations.** The adversary does not necessarily possess information about the deployed model nor the exact method to produce the accompanying explanations. Therefore, it is crucial for systems to be as resistant to transfer attacks as possible in order to evade perturbations constructed on similar models and explanations.

Thus, we examine how our classifiers and attribution methods react to transfer attacks computed with TEF. We alter the input samples for a given model and explanation method with TEF, then evaluate the PCC of attributions on the same samples but different architectures and explainers. The results are found in Figure 5. We observe that transfer attacks perform better than RA, some even by approx. $0.4$ in terms of average PCC decrease in the operation area of $\rho \approx 0.1$. However, as expected, they significantly fall short of the performance of TEF. Therefore, we conclude that transferring TEF perturbations across models and explainers effectively highlights fragility of explanations, but TEF provides tighter AR bounds without transfer.

**Semi-universal perturbations.** In this section, we take a step towards defining universal perturbations, similarly to the work of the authors Moosavi-Dezfooli et al. (2017) and Gao & Oates (2019). These provide fast and computationally cheap perturbations during attack time that are able to mislead classifiers with pre-computed perturbations. However, we attack the explanations of text classifiers, instead of their predictions and call our perturbations *semi-universal attack policies*.

We split the test dataset into two equally sized parts, the attack set and the evaluation set. We utilize the former for constructing our *semi-universal attack policies* and the latter to evaluate how effectively our semi-universal attack alters the attributions maps of our models.

First, for each sample in the attack set, we compute the optimal TEF perturbation for all our models and explainers. We then extract statistics of these perturbations, which are the most common replacement and the replacement frequency for each replaced token and sort these by decreasing frequency. These are our *semi-universal attack policies*, seen in Figure 6. During this *construction*

| AG's News | | | IMDB | | | MR | | |
|---|---|---|---|---|---|---|---|---|
| Token | Repl. # | Replacement | Token | Repl. # | Replacement | Token | Repl. # | Replacement |
| reuters | 146k | goldman | movie | 430k | cinematographic | movie | 8.2k | cinematographic |
| said | 131k | avowed | film | 338k | cine | film | 8.0k | cinematographic |
| new | 130k | nouvelle | good | 122k | decent | story | 2.6k | conte |
| ap | 107k | ha | great | 103k | whopping | good | 2.5k | decent |
| oil | 72.8k | tar | bad | 102k | wicked | comedy | 2.4k | humorist |
| ... | ... | ... | ... | ... | ... | ... | ... | ... |
| workers | 10.9k | labourers | amazing | 17.1k | staggering | triumph | 139 | victory |
| ... | ... | ... | ... | ... | ... | ... | ... | ... |
| zone | 2.9k | field | scary | 6.8k | fearful | shines | 69 | glows |

Figure 6: Semi-universal attack policies for different datasets.

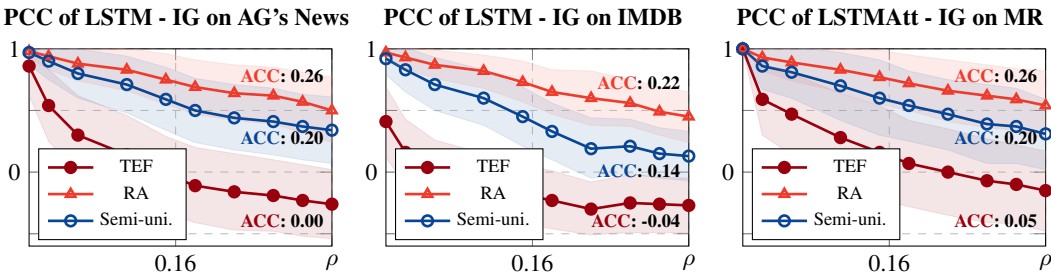

Figure 7: Average PCC of the indicated architectures and explainers after applying the semi-universal perturbations (Semi-uni.), compared to TEF and RA attacks. The semi-universal attack successfully decreases the correlation of original and attacked attribution maps.

phase, we query the model for predictions and explanations, as we compute optimal TEF perturbations.

Second, we *evaluate* our semi-universal attack that utilizes the aforementioned policies to alter explanations of classifiers. The inputs to this attack are a text sample, a semi-universal policy and a maximum perturbed ratio $\rho_{max}$. The attack iterates over the policy, starting with the token in the first row and finishing with the last. Whenever the current token is found in the input text sample, it is replaced with the replacement token in the list. If the perturbed ratio exceeds $\rho_{max}$, the attack is aborted. In such a way, perturbed inputs are created without querying the model during attack time. The actual perturbation for each text sample depends on the sample, hence the name *semi-universal attack policy*. The resulting samples are then evaluated on a given model and explanation method. Representative results are given in Figure 7. We conclude that our semi-universal policies are effective in reducing attribution correlation when the adversary has no access to the target model and explanation method, as indicated by the lower ACC values of the semi-universal PCC curves.

## 5 CONCLUSION

In this work, we introduced a novel black-box attack called TEXTEXPLANATIONFOOLER, that successfully perturbs input data such that the outcome of popular explanation methods in sequence classification, but not the prediction of the classifier. This attack provides a baseline estimator for attribution robustness and highlights the lack of robustness of current explanation methods. We compared it to the random attack, showing its superior performance to it on five different, widely used text classification datasets. Moreover, the transfer capabilities of the attack are evaluated. Finally, we showed the existence of semi-universal perturbation policies that are capable of altering explanations without querying the model during attack-time, even without having access to perturbations for those models. In future work, we plan to examine whether a similar white-box attack that has access to model gradients can improve robustness estimation. Moreover, instead of synonym embeddings, we plan to use BERT-based masked language models to extract possible candidates, further improving imperceptible word substitutions.

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
