# OpenReview forum: "Fooling Explanations in Text Classifiers"
_ICLR.cc/2022/Conference — ICLR 2022 Poster_

### Official Review · Reviewer_p6VB · 2021-10-29

**Correctness:** 3
**Technical Novelty And Significance:** 3
**Empirical Novelty And Significance:** 3
**Recommendation:** 5
**Confidence:** 4

**Main Review:**

--- Strengths:
-	The manuscript is well written in general and overall methodology makes sense.
-	Decreased correlations are indeed a valuable finding in terms of the proposed attack.
-	Proposed work is connected quite nicely to the previous work.
-	Evaluations on several datasets and methods are a plus.

--- Weaknesses:
-	Even though the claim is evaluation on five datasets, only the subset of the results is reported. As not much related work is available in this domain claimed by the authors, I would expect somehow reporting complete set of results.
-	Despite extensive evaluation, a limited amount of discussion exists. Implications on unchanged predictions are also not discussed.
-	As RA utilizes a straightforward baseline, I think it is relatively straightforward to have lower correlations with the proposed method compared to RA.
-	Please report different perturbation ratios in the plots.

**Summary Of The Paper:**

The authors introduce a text explanation attack algorithm called “Text Explanation Fooler”, that changes output of explanation methods while keeping the classification output the same. This is done by maximizing the distance between attribution maps of original and perturbed sequences, while forcing these inputs have the same prediction outcome. In the proposed attack, the authors apply word importance ranking for substitutions, and take POS tags and stopping words into account. They evaluate their methods with AG’s News, IMDB, Fake News, MR, and Yelp datasets along with CNN, LSTM, LSTMAtt, and BERT models and report significant decrease in correlations compared to RandomAttack (RA).

**Summary Of The Review:**

While the technical methodology seems to be valid and confirmed with the experiments, there are some drawbacks such as not reporting all the results, limited discussion, and partly applying some methods from vision domain to text processing, also reported in the weaknesses field of the main review. Even though, I think the contribution is timely and valid, however, there should be several updates on the manuscript before it is ready for publication.

---

> ### Author Response · Authors · 2021-11-16
> **Response**
>
> Thank you for pointing out valid weaknesses of our paper.
>
> As the space is limited in the manuscript, we only showed the most important results of our research. We will add an appendix with the complete set of results of our experiments to our final version. These results yield similar insights as the ones already included in the manuscript.
>
> With respect to the discussion and implications on unchanged predictions, we would like to point out the following. We view unchanged predictions as constraint for a “successful” attack. This is commonly done in the image domain, see referenced works in the paper. Intuitively, we argue that the users of such explanation systems might not be fooled had the classifier given different predictions to the altered explanations. They would just notice odd behavior, without trusting prediction nor explanation. Of course this is subject to discussion (little research has focused on this aspect of fragile attributions), yet we define our problem setting like this, similarly to other current work in the image domain.
>
> RA serves as a baseline for TEF that represents the “no knowledge” attack, similarly to random noise in the image domain. We use this as baseline for comparison, as such an attack is easy to construct. Indeed, we agree that it is expected for TEF to outperform RA, but as there is no other attack to compare our results to, this is the best we can do.
>
> The different perturbation ratios are already reported in the x-axes of the plots. They contain the ratio of changed tokens to the overall number of tokens in each sample. We emphasize that the values on the x-axes are different from the parameter rho of TEF, as the parameter of TEF contains the maximal ratio of perturbation after which the TEF is aborted. We will change the name of the parameter to rho_max in the final version to avoid confusion and make this distinction clearer.
>
> Thanks a lot for the constructive feedback and looking forward to hearing your insights on these points.
>
> Main changes planned for manuscript:
> - Add appendix with results on attribution robustness
> - Include more results on transfer attacks and universal perturbations
> - Better emphasize how RA serves as baseline
> - Change parameter name in TEF algorithm
> - Extend results section with more discussion on implications of fragile explanations in NLP, add Section on unchanged predictions (subject to space restrictions)

---

### Official Review · Reviewer_ZvQi · 2021-11-03

**Correctness:** 3
**Technical Novelty And Significance:** 3
**Empirical Novelty And Significance:** 3
**Recommendation:** 6
**Confidence:** 4

**Details Of Ethics Concerns:**

I do not find any ethics concern with this paper

**Main Review:**

Strength:
1. The authors provide good explanations in the background of the issue. I find section 2 and 3 generally well-written and some discussions and intuitions were given about the attack algorithm.

2. The method proposed is novel. The authors follow Ghorbani et al 2019 in looking at the issue on perturbation to explanations and extend the problem to text (discrete input). I believe the robustness of model's explanations is not a very well-studied issue in NLP and the community can benefit from this paper.

Weakness:
1. My first concern is that I am confused by section 4.3. I fail to see how the modified attack method provides universal perturbations to text explanations (without querying the model). I think this section would be clearer if the authors could use stick with their notations in section 2, 3 and algorithm 1 and provide further explanations.

2. Another concern is that I wish the authors could have made it clearer how finding s_adv (eqn 6) gives an attribution robustness estimation. I think many practitioners could benefit from a quantitative measure on the robustness of explanations.

3. Would the authors clarify an unreferenced $\tilde{W}$ in eqn 4?

4. One minor issue is that in investigating the attribution robustness in NLP models, the authors actually only studied sequence classification tasks (as opposed to span-labelling, text generation ...). I think this is a fine scope but also believe that the authors should perhaps make that clearer.

Look forward to the discussions with the authors/other reviewers and I am willing to adjust my scores if the authors could clarify weakness 1 and 2.

**Summary Of The Paper:**

This paper focuses on the issue of assessing the robustness of the "explanations" for NLP models (how the explanation, not the prediction, for NLP models, change with respect to small perturbations in the input). Particularly, the authors

1. formalized the problem of attribution (explanation) robustness estimation as an optimization tasks in eqn (4, 5, 6), which involves finding the perturbations that keep the model's output the same, while maximizing the difference in attribution (explanation) compared to the case without these perturbations.

2. developed a black-box attack model that estimate AR, without requiring gradient information. The authors also further conducted ablation studies on the components of the black-box model.

3. conducted evaluations of the authors' attack model, analyzing three existing popular attribution estimation methods (saliency maps, integrated gradients and attention as explanation)



**Summary Of The Review:**

Novel paper investigating attribution (explanation) robustness in NLP models. Some sections read confusing. I recommend "weak accept".

---

> ### Author Response · Authors · 2021-11-16
> **Response**
>
> We appreciate the positive feedback, thank you very much.
>
> With respect to our semi-universal perturbations (1), we hope to clarify things with the following points. First, in the referenced works, a universal perturbation is a single precomputed noise pattern that is added to the samples that are attacked, without querying the model. In our case, this is a perturbation policy that is applied to the sample, without querying the model. These are the priority lists described in section 4.3. The actual noise pattern depends on whether the tokens in this policy are present in the sample or not. Hence the name semi-universal. We call this the attack time.
>
> Second, both the referenced universal perturbations and our policies require pre-computation of some kind of perturbations on some samples with a non-universal attack. This is when the underlying classification model actually gets queried. In our case, this is TEF on the attack set. Based on these perturbations, the universal patterns are constructed. We call this construction time. Therefore, the universal perturbations do not query the model during attack time, only during construction time. We will modify the manuscript to make these points clearer in the final version.
>
> With respect to (2), there is indeed some confusion. In the image domain, it is straightforward, as attribution robustness is usually measured in the maximal change of attributions with the constraint that the altered input needs to be within a small epsilon-bound L_inf neighbourhood of the original image. This constraint is generally used to make sure changes are imperceptible and do not change the underlying label of the data. In our case though, as input is discrete, we can not simply apply this constraint. Therefore, we search for the input that maximizes attribution distance with linguistic constraints. However, these linguistic constraint tend to vary across different NLP attacks. We make sure our changes are imperceptible by constructing them from synonyms of the substituted tokens and applying linguistic filters. This is difficult to capture in an Equation, therefore Equation 6 is our only mathematical formulation of attribution robustness. Would you have any suggestion on how to better capture this?
>
> Moreover, thank you for pointing out the unreferenced W in Equation 4 and we will gladly emphasize stronger that we only consider sequence classification problems. We believe extending the problem to other tasks like text generation requires more thought and consider it future research.
>
> Thank you again for the good feedback.
>
> Main changes planned for manuscript:
> - Extend description of semi-universal perturbations in Section 4.3
> - Describe how s_adv in Section 3.1 gives robustness estimation, similarly to above
> - Scope problem set better and fix typos, unreferenced variables and some formulations

---

> > ### Comment · Reviewer_ZvQi · 2021-11-24
> > **Thanks for the detailed response**
> >
> > Thank you for the detailed response and I think the to-do list looks great.
> >
> > Regarding my original "weakness (2)", I think it is clearer now that s_adv is the solution to eqn (6), within certain constraints, and for text settings, these constraints are harder to capture. As a suggestion on how to illustrate this point more clearly, I think the authors may add an example that has: 1) the task, 2) the original input s, 3) the constraint applied, 4) a few examples in the neighborhood of s and 5) the final solution s_adv. I believe a figure or even just some text dump in the appendix could help clarify that point.
> >
> > Additionally, I am still a little confused as to how s_adv will give a robustness measure. Are the authors arguing that the difficulty of finding s_adv (or the distance between s_adv and s) gives a measure on robustness? It'd be helpful if the authors could explain further on that too.
> >
> > Thanks.

---

> > > ### Author Response · Authors · 2021-11-27
> > > **Answer to clarify questions better**
> > >
> > > Thank you for the clarification and the suggestion. Indeed, an example would help. As the modification deadline unfortunately is over, we will add something similar to what you suggested in the final version of the appendix, in case of an accept.
> > > With respect to the robustness measure, we define attribution robustness (AR) for a given input sample s as in Equation 4. As such, the robustness is inversely proportional to the maximal distance between the attribution map of the original sample and the attribution map of any input sample within the neighborhood of the original, fulfilling the aforementioned linguistic and prediction constraints. Therefore, the adversarial input s_adv does not directly give a measure of robustness, nor does the distance (in input embedding space) between the original sample s and s_adv. It is the distance between the attribution maps A(s) and A(s_adv) that these inputs induce that gives the measure of robustness. TEF finds an input that (locally) maximizes the this distance. If it returns an input text sample that induces an attribution map "very dissimilar" to the attribution map A(s) induced by the original sample s, the robustness is low. If TEF returns an input whose attribution map is "quite similar" to the original, the robustness is high. This is captured by Equation 6, whose solution finds a sample that maximizes exactly this distance, therefore is an estimator for AR, for a given input sample s. Then, the robustness for the whole model is calculated as the expected robustness for over each sample of the dataset.
> > > We hope this helps as clarification, and are happy to answer any further questions.

---

> > > > ### Comment · Reviewer_ZvQi · 2021-11-27
> > > > **Authors clarified my concerns effectively. I recommend accepting this paper.**
> > > >
> > > > Thanks again for the detailed response!
> > > >
> > > > The authors' response clarified my confusion as to how the solution to equation (6) (s_adv) gives a robustness measure that practitioners can rely on. I also recommend that the authors make revision to the final paper by integrating the explanations in their response above. The authors' clarification was very helpful.
> > > >
> > > > Overall, I recommend accepting the paper. If the authors could make all the revisions promised, I think this paper could be very helpful to the NLP community interested in explainability issues.

---

### Official Review · Reviewer_ZcSr · 2021-11-05

**Correctness:** 3
**Technical Novelty And Significance:** 2
**Empirical Novelty And Significance:** 3
**Recommendation:** 5
**Confidence:** 4

**Main Review:**


The proposed task is very interesting and important. Attacking explanations of deep neural networks have already been explored in the image domain [1,2] and graph domain (GNNs)[3]. It's interesting to have the investigation on NLP task.


The proposed attacking method is a general framework for attacking NLP model [4,5]. In particular, they compute the importance score for words. And then search the substitution to replace the words.


The proposed attribution robustness sounds nice in Eq. 4 and 6.
Although they are almost the first work to attack explanations in NLP task (???), the solution is not too novel and straightforward. In fact, they just change the objective function of attackers to achieve their goal.

 So most existing attacking methods in NLP can be applied to attack explanation of text classification.
In this case, the solutions to attack the explanation of text classification would be trivial. Are there any advanced attacking methods for this problem?

Minior: Chapter 2 -> section 2.




[1] Fooling Network Interpretation in Image Classification

[2] Fooling Neural Network Interpretations via Adversarial Model Manipulation

[3] Jointly Attacking Graph Neural Network and its Explanations

[4] Is BERT Really Robust? A Strong Baseline for Natural Language Attack on Text Classification and Entailment

[5] Adversarial Attacks and Defense on Texts: A Survey


**Summary Of The Paper:**

This paper proposes an attacking method to attack the explanations of text classification in NLP. The proposed method is easy to follow.
In particular, they first do the word importance ranking for each token of the input sample. High-important words are prioritized during substitution.  Second, candidate selection via counter-fitted GloVe is proposed to replace the word to attack explanations while maintaining the same confidence in the correctly predicted class. The authors conducted comprehensive experiments to evaluate the effectiveness of their methods.


**Summary Of The Review:**

The proposed task is very interesting and important. Attacking explanations of deep neural networks have already been explored in the image domain [1,2] and graph domain (GNNs)[3]. It's interesting to have the investigation on the NLP task.

---

> ### Author Response · Authors · 2021-11-16
> **Response**
>
> Thank you very much for the constructive feedback.
>
> Our attack provides a strong first baseline in estimating attribution robustness in NLP, which is novel and has not been established yet. While we do agree that it is similar to other work in a few aspects, we believe that having this baseline itself has plenty of added value, for the following reasons.
> First, we bring insight that, even with such adaptations of already existing attacks, we are able to significantly alter attributions, with resulting PCCs of around 0.0, see examples. To us, this strongly suggests that the research direction is worth exploring further.
> Second, our research community can build on it in order to improve attribution robustness estimation in NLP and track progress in this field. We hope that by establishing such a reference work, we encourage other researchers to develop more advanced attacks that give tighter estimations for attribution robustness.
>
> We hope you can somewhat agree with the points we described and that this helps to clarify why we chose our attack. We are looking forward to hearing your opinion on these aspects.

---

### Decision · Program_Chairs · 2022-01-20

**Decision:**

Accept (Poster)

**Comment:**

This paper presents an "attack"—TextExplanationFooler (TEF)—that adversarially (minimally) edits inputs such that the resultant attribution assigned by common explanation methods changes substantially, while the prediction does not. This is an extension of methods and results in vision to NLP. The authors find that all methods are vulnerable to this attack.

Reviewers agreed that the demonstration that perturbation attacks used in vision are applicable in NLP is interesting and may lead to follow-up work. The paper would benefit from technical clarifications in several places (see R2), which were largely resolved in discussion.